# Comparative Analysis of the Genetic Composition of Minorities in the Carpathian Basin Through Genome-Wide Autosomal Data

**DOI:** 10.3390/genes16050607

**Published:** 2025-05-21

**Authors:** András Szabó, Katalin Sümegi, Zsolt Bánfai, Kinga Hadzsiev, Ferenc Gallyas, Attila Miseta, Miklós Kásler, Béla Melegh

**Affiliations:** 1Department of Medical Genetics, Medical School, Clinical Centre, University of Pécs, Szigeti út 12., 7624 Pécs, Hungary; sumegi.katalin@pte.hu (K.S.); banfai.zsolt@pte.hu (Z.B.); hadzsiev.kinga@pte.hu (K.H.); melegh.bela@pte.hu (B.M.); 2Department of Biochemistry and Medical Chemistry, Medical School, University of Pécs, Szigeti út 12., 7624 Pécs, Hungary; gallyas.ferenc@pte.hu; 3Department of Laboratory Medicine, Medical School, University of Pécs, Ifjúság út 13., 7624, Pécs, Hungary; attila.miseta@aok.pte.hu; 4Institute of Hungarian Research, Andrássy út 64., 1062 Budapest, Hungary; mkasler10@icloud.com

**Keywords:** Carpathian Basin, ethnic minorities, population genetics, genome-wide marker data, allele frequency, haplotype analysis

## Abstract

Background/Objectives: The Carpathian Basin is a genetically and culturally diverse region shaped by complex historical migrations and various ethnic groups. While studies based on Y-chromosomal and mitochondrial DNA have provided valuable insights into the genetic diversity of these populations, genome-wide autosomal SNP data remain underutilized in understanding the genetic structure of these groups. This study presents the first genome-wide autosomal SNP-based analysis of key Hungarian-speaking ethnic groups in the region, focusing on admixture patterns and the extent of preserved historical genetic components. Methods: We analyzed genome-wide autosomal SNP data from 597 individuals representing several ethnic groups in the Carpathian Basin. Standard population genetic methods were applied to assess genetic structure, admixture and differentiation, with comparisons to broader European reference populations. Results: Most ethnic groups displayed genetic affinities with Eastern European populations, consistent with historical and geographical proximity. The Swabian group, of German descent, exhibited a distinct Western European genetic component, likely due to historical isolation. Transylvanian populations appeared relatively homogeneous, indicating a shared ancestral background. In contrast, Csangos showed distinct sub-clusters, suggesting population isolation and distinct histories. Overall, genetic homogeneity characterizes the region, though certain isolated groups retain distinct ancestral signatures. Conclusions: Autosomal SNP analysis revealed mild overall genetic structuring among Carpathian Basin ethnic groups. However, historical isolation has preserved unique genetic components in specific groups, highlighting the value of genome-wide data in uncovering fine-scale population structure. These findings contribute to a deeper understanding of regional genetic diversity, which has implications for both population history and health-related genetic research.

## 1. Introduction

The Pannonian Basin, or Carpathian Basin, is a large topographically discrete intermountain unit set in the Central European landscape, surrounded by the Carpathian Mountains, the Alps and the Dinaric Alps and divided roughly in half by rivers Danube and Tisza. The Carpathian Basin comprises distinct geological regions, including the Carpathian Mountains with rugged terrains, the Pannonian Basin characterized by low-lying sedimentary deposits, the Transylvanian Basin with basin and range topography and the Alpine-Carpathian Foreland featuring foothills and foreland basins [1,2,3].

The geographical landscape units of the Carpathian Basin do not always correspond to political state borders but may belong to several countries. According to current state borders, the Pannonian Basin centers on the territory of Hungary, which lies entirely within the basin, but it also includes Slovakia, southern Czech Republic, southeast Poland, southwest Ukraine (Transcarpathia), western Romania (Transylvania), northern Serbia (Vojvodina), northern Bosnia and Herzegovina, northeast Croatia (Drava-Sava region), northeast Slovenia (Mura Statistical Region) and eastern Austria (Burgenland).

The history of the Carpathian Basin is deeply intertwined with the movements of various immigrant populations throughout millennia. Archaeological evidence suggests that *Homo sapiens* first appeared in the region approximately 20,000 years ago, marking the beginning of human settlement. Over time, waves of different tribes migrated into the basin, including Celts, Romans, Goths, Huns and Slavs, each leaving their mark on the cultural landscape. The arrival of the Magyars in the late ninth century brought significant changes, establishing the foundations of the Hungarian state. Throughout its history, the basin served as a crossroads for trade and cultural exchange, facilitating the mingling of diverse ethnic groups and languages. The medieval period witnessed the rise of great kingdoms and the spread of Christianity, further shaping the social fabric of the region. The Ottoman conquest in the 16th century introduced yet another layer of cultural influence, particularly in the southern parts of the basin. Despite periods of conflict and conquest, the basin remained a melting pot of different ethnicities, with populations often intermixing and adopting elements from one another. The 19th and 20th centuries brought significant demographic shifts, with industrialization and urbanization leading to the influx of people from rural areas into cities. Today, the Carpathian Basin continues to be a region of rich cultural diversity, reflecting its long history of migration and interaction among various peoples [4,5,6].

In the Carpathian Basin, there are several Hungarian-speaking minority communities, predominantly inhabiting territories that were once part of the historical Kingdom of Hungary. These minorities are spread across several countries, including Hungary, Romania, Slovakia, Serbia, Ukraine and Croatia. While they have preserved their cultural and linguistic distinctiveness, these minorities have often undergone varying degrees of assimilation, which have influenced their identity and status within the region.

According to the 2011 census, the Carpathian Basin had 25,700,000 inhabitants, which is 1.1 million less than the previous estimation in 2001. The number of Hungarians is officially 8.32 million in Hungary, 1.23 million in Romania, 459,000 in Slovakia, 251,000 in Serbia, 141,000 in Ukraine (data from 2001), 14,000 in Croatia and an estimated 10,000 in Austria and 4000 in Slovenia. Approximately 1 million people did not declare their nationality in the last census, of which 500,000 live in Hungary, 400,000 in southern Slovakia and 100,000 in Vojvodina [7,8,9,10,11,12,13,14].

The examination of Y-chromosome haplogroups in Europe revealed the presence of several dominant haplogroups. Of these, haplogroups R1a and R1b showed the highest proportion, with Western European R1b and Eastern and Northern European R1a dominance [15,16,17,18,19,20]. In addition, the presence of I1a in Northern and Western Europe, I1b in Eastern and Southern Europe, N3 in Northern Europe and the high proportion of haplogroup J in Central and Southeastern Europe should be highlighted [20]. The distribution of Y-chromosome haplogroups in the Carpathian Basin forms the transition between Eastern and Western European countries, with an approximately equal share of dominant R1a and R1b. In addition, the high share of E3b, I1a, I1b and J characteristic of Central and Eastern European countries should be highlighted, indicating the region’s diversified genetic environment [21,22,23].

The examination of the mitochondrial haplogroups of Europe was also the subject of many previous studies, as a result of which a higher proportion of the haplogroups H, U, J, T K, V and W were found [24]. In terms of the Carpathian Basin region, the relatively high occurrence of H1, H3, H5, J, K, T1, T2, U4, U5, V and W should be highlighted, showing the dominance of H1 and H3, demonstrating the region’s complicated historical migrations and interconnections [25,26].

Previous studies focusing on the genetic background of Hungarian-speaking populations in the Carpathian Basin have revealed evidence of complex admixture processes. Genome-wide and lineage-based analyses have reported predominantly European ancestry with minor but detectable Central and Inner Asian components, most notably among modern Hungarians [27,28]. Regional investigations have also suggested population substructure within Transylvanian Hungarian-speaking groups, while large-scale admixture analyses have included Hungarian samples in broader Eurasian gene flow events [29,30]. Our earlier analyses further indicated signals of detectable East Asian/Siberian Turkic-related ancestry specifically among the Csango population, a finding that appears to be unique within Central and Eastern Europe and calls for further in-depth examination [31].

Our aim is to create a comprehensive picture of the population conditions of the Carpathian Basin including the main Hungarian-speaking minorities, using genome-wide autosomal single nucleotide polymorphism data. Our objective was to examine whether the individual ethnic groups are genetically distinct from each other, whether genetic traits linked to their original ancestry can still be identified and to what extent the Carpathian Basin constitutes a genetically homogeneous unit from a population genetics perspective. These investigations might provide a basis for the developing fields of personalized medicine and pharmacogenetics, as they offer crucial insights into the genetic composition of Hungarian-speaking minorities in the Carpathian Basin. Understanding the genetic structure of these populations is critical for identifying disease susceptibilities and optimizing drug responses, which are central to personalized healthcare. With the increasing focus on tailoring treatments to individual genetic profiles, such research is key to developing more effective and safer therapies, specifically tailored to the unique genetic characteristics of these communities.

## 2. Materials and Methods

### 2.1. Datasets

We collected and genotyped 98 Hungarians from Hungary; 41 Hungarians from the historical region of “Felvidék”, Slovakia; 61 Danube Swabian individuals from Southwest Hungary (Dunaszekcső area) and 48 Transylvanian Hungarian (24 Hungarians from settlements dating back to the era of the Arpad Dynasty and 24 Hungarians randomly collected from Transylvania), 266 Sekler (26 from Bukovina, 58 from Gyergyó, 30 from Havad, 34 from Kézdivásárhely, 25 from Nyárádmente, 17 from Székelykocsárd, 27 from Székelyudvarhely, 24 Seklers from Korond and 25 Seklers randomly collected from Transylvania), 83 Csango (35 Moldavian Csango and 48 Gyimes Csango) individuals on the Illumina Infinium Global Screening Assay. Ethnicity of the sample donors were based on self-declaration. The recruitment period took place between 1 September 2022 and 31 March 2024. Those were included in our investigations, whose ancestry does not include ancestors belonging to other ethnicities going back at least three generations, according their established pedigree (Figure 1).

Genotyping was performed by a third-party service provider using the Illumina Infinium Global Screening Array v3.0 BeadChip, which includes approximately 650,000 SNPs. DNA was extracted from leukocytes obtained from EDTA-anticoagulated blood samples. We performed quality control procedures using the Illumina GenomeStudio 2.0 software and the PLINK1.9 package to ensure the integrity of the dataset [32,33]. Hardy–Weinberg Equilibrium (HWE) filtering was performed separately for each population; for groups with fewer than 50 individuals, we applied a *p*-value threshold of 1 × 10^−3^, while for larger populations, we used 1 × 10^−6^, acknowledging that deviations from HWE may reflect demographic structure rather than technical error. SNPs with a Minor Allele Frequency (MAF) below 0.05 were excluded using PLINK1.9 ‘maf’ flag. Individuals with more than 0.05 missing genotype rate were removed using the ‘mind’ function of PLINK, although no such cases were found. Using the ‘geno’ flag in PLINK1.9 with a missing call rate threshold set to 0.1, SNPs with more than 10% missing data were also excluded from the data. Genetic distances were added to the marker data using the HapMap Phase 2 GRCh37 genetic map [34]. After all filtering steps, the final dataset comprised 90,503 high-quality SNPs.

Before participation, each individual received verbal information about the study and provided written informed consent. The study was approved by the Regional Research Ethics Committee of Pécs, and all samples were anonymized. Ethical procedures followed the principles of the Declaration of Helsinki.

As a collective term for the investigated populations, we used the abbreviation HSE (as Hungarian-speaking ethnicities) throughout the article. The following groups were created from the collected and genotyped data of these investigated populations: non-Sekler Hungarians living in Transylvania (TransHun); Transylvanian Hungarian population living in settlements established in the era of the Árpád Dynasty, 1000–1301 AD (ArpadHun); nine Sekler groups from the areas of Korond (KorondSekler), Székelyudvarhely (SzekelyudvSekler), Székelykocsárd (SzekelykocsSekler), Nyárádmente (NyaradmenteSekler), Kézdivásárhely (KezdivasSekler), Havad (HavadSekler), Gyergyó (GyergyoSekler) and Bukovina (BukovinaSekler); Seklers from various areas throughout Transylvania (TransSekler); Moldavian and Gyimes Csangos (MoldavianCsango, GyimesCsango); Slovakia-living Hungarians (SlovakHun); and Danube Swabian people living in Southwestern Hungary (Swabian).

Additional ethnic groups included in this study were collected from various publicly available datasets. These public data were the CEPH-Human Genome Diversity Project data (HGDP European samples, *n* = 160, from eight populations (Adygei, French Basque, French, North Italian, Orcadian, Russian, Sardinian, Tuscan), and HGDP Asian samples, *n* = 207, from nine populations (Balochi, Brahui, Burusho, Hazara, Kalash, Makrani, Pashtun, Sindhi, Uyghur), genotyped on an Illumina 650Y array); various datasets from the public repository of the Estonian Biocentre (German, *n* = 10, and Romanian, *n* = 14, genotyped on different Illumina genotyping arrays); and European populations from the 1000 Genomes Project (1KGP, *n* = 509 from five populations (CEU, FIN, GBR, IBS, TSI), genotyped on an Illumina InfiniumOmni2.5–8 array [35,36,37,38,39,40,41,42]. HGDP data, datasets from the Estonian Biocentre and 1KGP data were applied widely in all investigations and represented the Carpathian Basin. These datasets were applied to study the European ancestry of the main minorities of the Carpathian basins.

As we were unable to collect samples from Serbia, one of the significant Hungarian-speaking minorities, the Hungarian population of Vojvodina was excluded from the analysis. The Romani population was also excluded from this study, as their unique South Asian origin requires a completely different approach and discussion. Additionally, the origin, genetic characteristics and history of the Romani have been extensively covered in numerous population genetics studies over the past decade, utilizing both sex chromosome-based and autosomal data analysis methods.

### 2.2. Inferring Population Structure and Ancestry

To explore the genetic relationships among Hungarian, Csango, Sekler, SlovakHun, Swabian and Transylvanian Hungarian (TransHun and ArpadHun) populations and the considered publicly available reference groups listed in the previous section, we employed three complementary approaches. Principal component analysis (PCA) was conducted using SMARTPCA, part of the EIGENSOFT 6.01 software package [43]. We tested the significance of the obtained principal components applying the built-in Tracy–Widom statistics function of SMARTPCA and calculated the explained variance for each significant principal components as well. Additionally, fixation index (F_ST_) values, representing average pairwise allele frequency differentiation, were computed with the same software. To assess the distribution of ancestral components across the studied ethnic groups in a perspective of a number of common hypothetical ancestral groups the STRUCTURE-like maximum likelihood estimation-based (ML) clustering method, implemented in ADMIXTURE 1.22 was also used [44]. Cross-validation (CV) error check was conducted parallel to the estimation process of ADMIXTURE in order to ensure the use of the correct number of clusters. We also investigated the population splits, admixture events and inferred migration processes that occurred in the past by computing an ML tree of the target populations based on genome-wide allele frequency data using the TreeMix 1.13 software [45]. Based on the criteria described below, we created two separate datasets for PCA and ADMIXTURE analyses. The first dataset contained the Hungarians, HSE populations (ArpadHun, TransHun, Seklers, Csangos, SlovakHun and Swabians) and European 1KGP populations (CEU, FIN, GBR, IBS and TSI). To minimize the impact of background linkage disequilibrium (LD) on both PCA and maximum likelihood-based ancestry analyses, LD-based pruning was performed using PLINK 1.9. The pairwise genotypic correlation threshold (r^2^) was set to 0.3, with a sliding window of 50 SNPs and a step size of five SNPs. This choice was based on both empirical testing and prior literature recommendations. Although the commonly used default is r^2^ = 0.2, we opted for the slightly more relaxed threshold of 0.3 to retain a greater number of informative SNPs while still minimizing the confounding effects of background linkage disequilibrium [43,44]. The number of individuals in the first dataset was 1106 and the number of SNPs after the LD-based pruning was 78,764. The second dataset contained the Hungarian and HSE samples with German and Romanian samples from the Estonian Biocentre data and HGDP European (Adygei, French Basque, French, North Italian, Orcadian, Russian, Sardinian, Tuscan) and HGDP Asian (Balochi, Brahui, Burusho, Hazara, Kalash, Makrani, Pashtun, Sindhi, Uyghur) samples. The pruned dataset with the Hungarian, HSE, Romanian, German, HGDP European and HGDP Asian samples counted 988 samples and 65,127 SNPs after LD-based pruning. For easier interpretation of the results and for better visibility, we plotted the results with and without HGDP European and German samples and HGDP Asian samples, and we divided the HSE group into subpopulations.

For the ML tree, we created a third dataset containing only our subject populations and only European reference populations. The dataset on which TreeMix analysis was performed contained the Hungarian, HSE, Romanian, German and HGDP European samples. With the exception of the Uyghurs, which was applied as the root population, Asian data from the dataset were removed for this analysis. The dataset contained n = 792 individuals and 65,127 SNPs. We applied a window size of 1000 SNPs and used the HGDP Uyghurs as the root population. According to the preliminary results and residual fit values, we did not incorporate any migration events in this analysis.

### 2.3. DNA Segment Analyses

Genome-wide identity-by-descent (IBD) estimation was carried out in order to investigate the genetic relatedness between the target populations including Hungarian, HSE, German, Romanian, HGDP European and HGDP Asian samples [46]. We also performed homozigosity-by-descent (HBD) analysis to investigate genome-wide autozygosity of the populations listed previously.

We employed the Refined IBD algorithm from Beagle 4.1 to detect IBD segments to further detail the distinctive characteristics among our subject populations. For the DNA segment analysis, we used the unpruned dataset applied in prior allele frequency-based investigations, which contained 90,503 SNPs. The major allele in the dataset was set as the A2 allele using PLINK1.9, and subsequently, it was converted to Variant Call Format (VCF) 4.1 using the PLINK/SEQ v0.10 conversion tool [47]. We specified a minimum IBD segment length of 3 cM, with an IBD trim parameter set to 10. Additionally, an IBD scale parameter was applied, following the √(n/100) recommendation, where n represents the dataset’s sample size and √(*n*/100) ≥ 2 [48]. All other parameters remained in their default settings. Average pairwise IBD sharing between populations was calculated as per the method detailed in the paper of Atzmon et al., 2010 [49].

The Refined IBD algorithm simultaneously identifies both IBD and homozygous-by-descent (HBD) segments, enabling inferences on genome-wide autozygosity across the studied populations. This method can indicate the extent of isolation and increased autozygosity within these groups, providing a basis for comparison with reference populations previously analyzed for genetic isolation. Consequently, we also calculated the average count and average total HBD segment length of detected HBD segments in each individual.

D-statistics were used to infer potential gene flow events and to investigate the degree of admixture between populations [50,51]. For these analyses, we employed the four-population test, which calculates the D-statistics based on allele frequencies across four populations. In our study, we included the Hungarian, Romanian, HSE, German and Swabian populations, with the Yoruba population serving as the outgroup. The D-statistic was calculated for each set of populations to identify significant deviations from zero, indicating potential introgression between the populations. We considered Z-scores above 2 as statistically significant, indicating an excess of allele sharing between the test populations. The results were used to assess historical population structure and gene flow, providing insight into the evolutionary relationships and genetic interactions among the studied groups.

## 3. Results

### 3.1. Relationship of the Hungarian, Slovakia Living Hungarian, Hungary Living Swabian and Transylvanian Samples to European Populations

We implemented principal component analysis (PCA) using SMARTPCA from the EIGENSOFT 6.01 package and the clustering software ADMIXTURE 1.22 to study the relationship of Hungarian and HSE populations to Europeans (1KGP European populations). The PCA results on eigenvectors 1 and 2 show that the Hungarian and HSE populations (except Swabians) cluster together, but a slight shift could be seen, because Hungarians are clustered in the direction of 1KGP CEU and GBR populations. The GyimesCsango and SlovakHun populations show very close but separate clustering, during which SlovakHun shows more of a Northern European orientation. The Swabian population shows a distinct, isolated clustering. In terms of its location, it shows a Western European clustering orientation, forming a transition between the localization of the Hungarian, Transylvanian, SlovakHun, CEU and GBR populations. As a result of the slightly elongated orientation of the Hungarian samples towards Western Europe, a few Hungarian samples and a few 1KGP European samples overlap with the localization of the Hungarian Swabians. The HSE samples are well separated from the Northern and Southern Europeans, thus representing the Central Eastern Europeans (Figure 2).

Common clustering of Hungarian and HSE populations is observable when represented on eigenvectors 3 and 4. This graph also shows the localization of the Hungarian samples slightly towards 1KGP CEU and GBR populations, while this could not be observed regarding the HSE populations. Despite the close clustering of the Hungarian, Transylvanian, SlovakHun and Swabian samples, it can be seen that the GyimesCsangos and the Swabians show separate clustering. The GyimesCsangos—but also the MoldavianCsangos—are closer to South Europeans on these eigenvectors, while the Swabians clustered more in the direction of the 1KGP CEU and GBR; therefore, the samples have North and West European ancestry, similar to the representation on eigenvectors 1 and 2 (Figure 2). Altogether, our samples show a very tight, solid grouping on the second two largest eigenvectors compared to 1KGP populations, which shows that these eigenvectors could capture much less diversity in our data and that our populations are genetically much more uniform than 1KGP European samples.

By comparing the subject populations with the HGDP and Estonian data, the appearance of the previous 1KGP results also can be seen. The Hungarian and HSE populations cluster together, overlapping mostly with German, Romanian, Orcadian and French populations, but are distinctly separated from the Russian, Adygei, Sardinian, French Basque, Tuscan and evidently from the HGDP Asian populations. However, a slight shift is visible between the Hungarian samples and the HSE samples (Figure 3). Comparing our samples with the HGDP data, in contrast to 1KGP, eigenvectors 3 and 4 show additional structure, since HSE samples are somewhat scattered on the fourth principal component. However, this loose clustering is supposedly caused by the South and Central Asian populations since seemingly they possess mainly a genetic variability on this eigenvector.

Investigating the subject populations separately and comparing them with the HGDP European and Germans and Romanians from the Estonian data, the results of eigenvectors 1 and 2 strengthen our previous findings. The subject populations are clustered together with the Romanians and West European populations like German, French and Orcadian populations. The SlovakHuns, Swabians and GyimesCsangos are separated from each other within the common cluster, as could be observed in the PCA carried out with the 1KGP data. Swabian patterns show an orientation to the French and French Basque samples, overlapping with Germanic patterns, thus representing their Western European origin. The GyimesCsango samples cluster in the direction of the of the Romanian samples, while the SlovakHun samples can be found in the opposite direction, mirroring their geographical relationship relative to our investigated populations (Figure 4).

Investigating the populations on the 3 and 4 eigenvectors, the common clustering of the Hungarian and HSE populations is also visible. However, it is clearly visible here as well that the GyimesCsangos show an elongated, separate clustering. In addition, similar to previous findings, the separation of the SlovakHun and Swabian samples within the common cluster can be seen here. While the SlovakHun samples are localized in the direction of the Eastern European Russian cluster, the clustering of the Swabian samples is more in the direction of the Orcadian and French populations, showing an overlap with the German samples. In addition, the clustering of the MoldavianCsango samples can be seen, which are relatively separate, close to the Romanians, clustering in the direction of the GyimesCsangos (Figure 4).

By plotting only the results of the subject populations and the Romanian samples, the separate clustering of the Swabian, SlovakHun and GyimesCsango samples on eigenvectors 1 and 2 can be observed. The Swabian and SlovakHun populations demonstrate clustering that shows a tendency of separation from the other HSE populations. They cluster towards different directions, and their clustering overlaps only with the Hungarian samples. The position of the Hungarian samples overlaps with almost all other samples, except for Romanians. The scattered plotting of GyimesCsango samples towards Romania can also be observed. This clustering pattern is a good representation of the actual geographical locations of these populations relative to each other, in which the Swabians represent the Western Europeans, the SlovakHuns the Northern Europeans, and the Hungarians and Transylvanian populations the Central Eastern European lines (Figure 5).

Separate clustering of the GyimesCsango, SlovakHun and Swabian individuals represented on the 3–4th principal components also can be seen; however, while the SlovakHun samples separate from the other samples mainly on the third principal component, Swabians and GyimesCsangos show significant separation on eigenvector 4. The MoldavianCsangos are also clustered in the direction of the GyimesCsangos, also showing a differentiation observed on eigenvector 4. The Hungarian samples also overlap with the Swabian, SlovakHun and other HSE samples, except for the MoldavianCsango and GyimesCsango samples. In addition, they mostly show clustering extended in the direction of the SlovakHun cluster. Transylvanian samples except GyimesCsangos are located between the localization of the Hungarian and Romanian samples, overlapping with both populations (Figure 5).

ADMIXTURE strengthens the PCA results. The cross-validation error check indicated the lowest CV-error in four hypothetical ancestral groups. However, European-derived clusters begin to separate at K = 5; therefore, the analysis result applying five clusters appears to be more informative, and the differences in corresponding CV-error values are negligible in the case of 3–5 hypothetical ancestral groups. The red cluster might indicate an ancient European ancestry that is more typical for the western regions of Europe and derived from the Neolithic period to a high extent, based on the observation that French Basque and Sardinian groups have the highest share from it. The blue hypothetical ancestral group might refer to a newer European, possibly Indo-European, or East European ancestry. The remaining three clusters are derived from Asia, where green might refer to the Ancestral North Indian genetic component, and the purple is possibly derived from Central Asia, while the yellow cluster represents a mixed South Asian lineage. At K = 5 hypothetical ancestral groups, the investigated populations appear to be similar, however, the Transylvanian populations show a slightly higher Central Asian share compared to Hungarian, SlovakHun and Swabian populations, which is unique among the populations of the Carpathian basin. The Swabian samples show the least Central Asian share and the highest share from the Western European component. The SlovakHun and Swabians, which exhibited isolated plotting on the PCA results, show a pattern more similar to the Hungarian and Transylvanian populations than to the HGDP European or Asian populations. Danube Swabians show the highest similarity with Germans on lower K values, but applying five hypothetical ancestral groups, the results show that the German samples are not homogeneous. Some German samples are from the eastern border of Germany, exhibiting a significant share from the blue cluster (Figure 6 and Appendix A).

On the ADMIXURE calculated using 1KGP data, at K = 2, the relative homogeneity and an equal share of the Hungarian and HSE samples from the European ancestral components are visible. In addition, it could be seen that the GyimesCsangos have a slightly higher share, while the SlovakHuns have a slightly lower share of the supposed South European (light blue) ancestral component. At K = 3, Swabians show a higher West European share compared to the Hungarian and other HSE populations (Appendix A).

The average pairwise allele frequency differentiation matrix (F_ST_ matrix) confirms the results of the allele-frequency-based ancestry estimation methods. Some Transylvanian groups like ArpadHun and SzekelykocsSekler, as well as MoldavianCsango and GyimesCsango populations, show higher F_ST_ values with all populations compared to other HSE populations (Figure 7, Appendix A).

The TreeMix graph captured the previously observed characteristics of populations and population relationships well. On the TreeMix graph, we can observe that isolated populations (Basques, Sardinians, Orcadians) and certain subpopulations from Transylvania (Seklers from Székelykocsárd, as well as GyimesCsangos and ArpadHuns) stand out from the Central European drift parameter (Figure 8 and Appendix A).

### 3.2. Haplotype-Based Analysis Results

According to the average pairwise IBD sharing results, the subject populations show a similar degree of shared IBD with Hungarians and Romanians. They also show a similarly high share with East (Russians) and East-Central European populations (Germans). Subject populations showed the lowest IBD share with the Caucasus region (Adygei), South Europeans (North Italians, Tuscans) and known isolated populations (Basques, Sardinians) (Figure 9). Average IBD share values of West Europeans (French, Orcadians) lie on the spectrum defined by the previously described lowest and highest sharing values.

The HBD analysis clearly shows the strong genetic isolation of the known closed populations (Basques, Sardinians) among the HGDP populations. Immediately after the known isolated populations, the Transylvanian populations appear in the figure, which show relative isolation (HavadSekler, ArpadHun, Moldavian and GyimesCsangos). In terms of genetic isolation, the other populations show the European average (Figure 10).

Most of the D-statistics analyses show that population relationships do not deviate from the expected unrooted phylogenetic tree. However, some of the results provide additional support for subtle yet notable genetic relationships among the studied populations. ArpadHun individuals exhibit a significantly closer genetic affinity to present-day Hungarians than to modern Romanians (D = 0.000303, Z = 2.316). Similarly, a few Sekler subpopulations, such as BukovinaSekler and HavadSekler, display modest but significant excess allele sharing with present-day Hungarians (D = 0.000262, Z = 2.122 and D = 0.000266, Z = 2.165, respectively). Most other Sekler and Csango groups do not show a clear directional bias, except that the MoldavianCsango subpopulation shows a closer relationship to the ArpadHun and GyimesCsango groups compared to present-day Romanians (D = 0.000323, Z = 2.445 and D = 0.000278, Z = 2.405, respectively). Additionally, the D-statistics setup investigating the ancestry of the Danubian Swabian population does not violate the expected unrooted phylogenetic tree, showing a more significant connection with the German reference samples (Table 1 and Appendix A).

## 4. Discussion

Our allele frequency-based methods revealed that the subject populations of our study are East-Central European populations with a stronger connection to Eastern Europeans, except for the Swabians, who have stronger West European ancestry.

In general, the Transylvanian groups show great homogeneity, except for Gyimes Csangos, as well as Moldavian Csangos. In the case of the two Csango groups, the reason for this genetic difference is presumably the same. As we stated earlier, one of our previous papers showed that Csangos exhibit an East Asian/Siberian Turkic genetic ancestry, which is unique in the East-Central European region. This can explain the phenomenon observed on the PCA figures in the case of Csangos (Figure 5).

A subtle structure can also be observed on the PCAs. Transylvanian populations show a closer clustering to Romanians, and the SlovakHun group plots close to the Hungarians, which mirrors the actual geographical location of these groups. The two poles of the Hungarian-speaking ethnicities are formed by SlovakHun and GyimesCsangos, and the orientations of the two poles are the Hungarian and Romanian clusters. The Swabians stand out noticeably from this arrangement due to their significant West European ancestry component.

DNA segment analyses revealed the main sources of ancestry, which were unsurprisingly the East-Central European populations. Another significant source of ancestry turned out to be East Europe. West Europe shows a lesser role in the ancestry of our Hungarian-speaking populations, but South Europe shows the least connection to them based on our IBD analyses. The only exception is the Swabians, who retain a significant amount of West European ancestry. According to our HBD analysis, the two Csangos (the HavadSekler and ArpadHun groups) show isolation comparable to known closed groups like the Basques and Sardinians, and all other Hungarian-speaking groups exhibit average European values, suggesting that they do not have a genetically closed society. The clustering of the two Csango groups, which sets them apart from the others, is somewhat explained by the HBD analysis. However, allele frequency-based analyses indicate that they have preserved different types of ancient traits in their genetic makeup compared to the Seklers.

The Hungarian-speaking ethnic groups of the Carpathian Basin exhibit strong homogeneity, which is slightly differentiated by their admixture with the respective local major populations, Hungarians, Romanians and—presumably—Slovakians. This also means that these groups do not show significant genetic isolation. An exception to this is the Csango people, who display both significant isolation and distinct ancestry compared to the other groups studied. Their isolation allowed for the preservation of their unique ancestry to a detectable degree. The origin of Transylvanian Hungarians, Seklers and Csangos is different. The settlement of Hungarians in Transylvania took place over several centuries, primarily from the late 10th to the 13th century. However, the precise timeline is still debated among historians, as artifacts of Hungarian origin dating to the first half of the 10th century have also been discovered in the region. This gradual migration and settlement led to the formation of distinct Hungarian communities in Transylvania, which can be traced back to the period of the Árpád dynasty. Meanwhile, the origins of the Székely people—known in English and German as the S(z)eklers and in Romanian as Secui—remain a subject of ongoing debate among scholars, with no definitive consensus yet established. Traditionally, they originate themselves from the Huns, but there are hypotheses also including an ancient Hungarian origin [52]. The first written evidence mentioning the Sekler people in Transylvania originates from the 12th century. The Seklers are one of the largest minorities of Europe, of which the majority (500–700,000 people) lives in Székely Land, Transylvania [53]. The origin of Csangos is also a matter of debate and there are many hypotheses regarding it. Nowadays, the most accepted theories are that Moldavian Csangos are either came from the territory of the Kingdom of Hungary or they are migrating descendants of the Sekler people who live nowadays in relatively strong isolation in an area inhabited mainly by Romanians. According to scholars, Gyimes Csangos are the descendants of Seklers and Moldavian Csangos migrating to the Gyimes area [54].

The Danube Swabians, an ethnic group that settled in the Carpathian Basin mainly during the 17–18th century, originated primarily from Southwestern Germany, such as Swabia, a cultural, historical region of Germany, and neighboring regions such as Alsace-Lorraine, belonging now to France and a former territory of the German Empire. Unlike other Hungarian-speaking ethnic groups in the area, who, according to our investigations, often display genetic affinities with Eastern European and Central Asian populations, the Danube Swabians retain genetic characteristics indicative of Western European ancestry. This distinct lineage likely stems from their relatively recent migration from Western Europe, preserving genetic differences that set them apart from other Hungarian-speaking populations in the region.

The D-statistics results are in line with the PCA and ADMIXTURE analyses, strengthening the overall patterns of genetic structure observed in the studied populations. Most comparisons are consistent with the expected unrooted population topology, showing no substantial deviations. Nevertheless, certain outcomes support the presence of subtle but relevant genetic relationships. The ArpadHun individuals show a slightly stronger genetic connection to present-day Hungarians than to modern Romanians, suggesting some degree of historical continuity or deeper shared ancestry along the Hungarian lineage. Similarly, minor but statistically significant signals of excess allele sharing are found in specific Sekler subgroups, such as the BukovinaSekler and HavadSekler groups, which may point to historical gene flow or regional continuity with the Hungarian population. Some of these findings are also consistent with the results of the HBD analysis. The ArpadHun and HavadSekler groups showed the highest degree of isolation from the investigated Hungarian-speaking populations. This limitation of the gene flow might have helped in retaining much of their ancestral Hungarian genetic heritage. However, the contemporary BukovinaSekler population does not show strongly restricted gene flow, and the D-statistics might capture the result of their more significant isolation in the past. In contrast, most other Sekler and Csango groups show no clear directional bias, implying a broadly homogeneous genetic background shaped by some degree of probable admixture with the local population. An exception is the Moldavian Csango group, which demonstrates increased allele sharing with both the ArpadHun and GyimesCsango populations compared to modern Romanians, further highlighting their unique genetic position among Hungarian-speaking communities and their relatively strong genetic isolation. Additionally, the Danube Swabians do not exhibit elevated genetic affinity with present-day Hungarians, in line with the preservation of their distinct Western European (German) ancestry, likely maintained through endogamy and relative isolation in the past.

Although the Hungarian-speaking ethnic groups of the Carpathian Basin exhibit strong homogeneity, there are also slight genetic differences between them, highlighting the region’s genetic heterogeneity. This makes pharmacogenetics studies highly relevant, particularly for populations that show higher isolation. They have retained a significant amount of their original genetic heritage and may also have undergone genetic drift, such as the investigated Csango groups and the Swabians. These groups may have distinct pharmacogenetic profiles and different susceptibilities to rare diseases. In this context, personalized medicine becomes not only important but also highly relevant, as it allows for more precise and effective healthcare tailored to the unique genetic characteristics of these populations.

## Figures and Tables

**Figure 1 genes-16-00607-f001:**
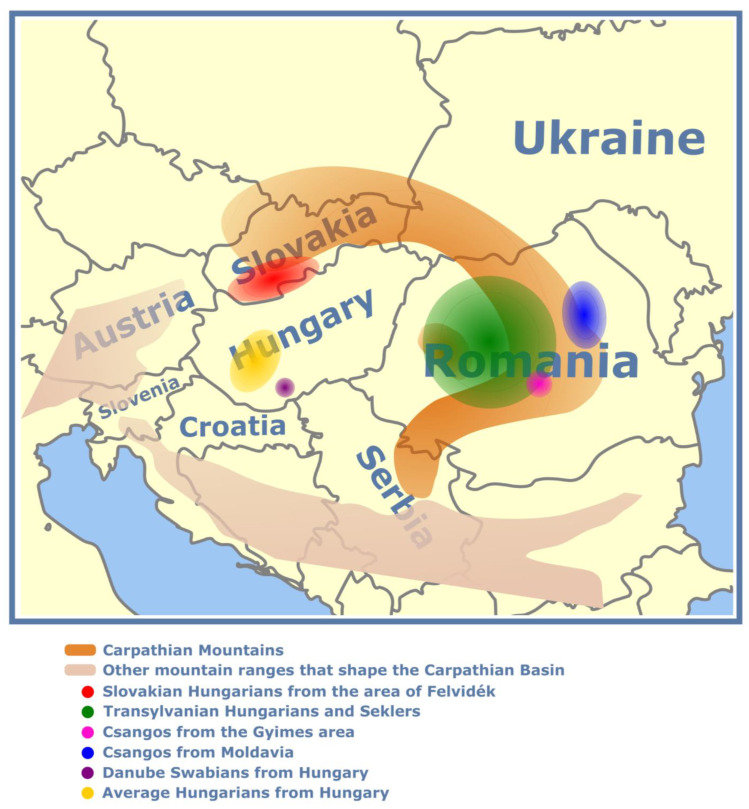
Schematic location of the investigated minorities in the Carpathian Basin.

**Figure 2 genes-16-00607-f002:**
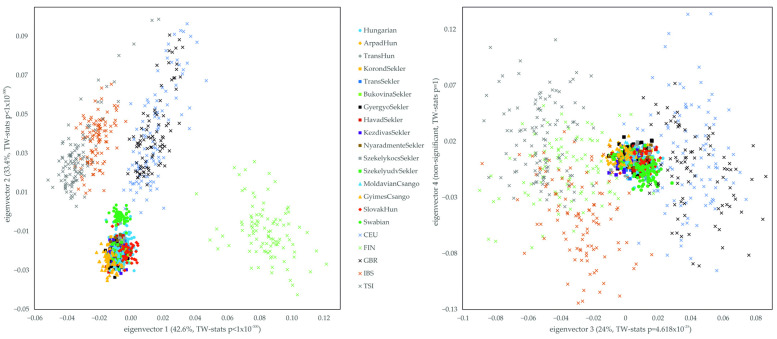
PCA analysis of the Hungarians, Slovakia-living Hungarians, Hungary-living Swabians, Transylvanian samples and various 1KGP European reference populations. Eigenvalues of eigenvectors 1, 2, 3 and 4 were 3.528, 2.767, 1.991 and 1.756, respectively. The statistical significance and the explained variance of each principal components are provided in Appendix A. Each symbol represents one individual.

**Figure 3 genes-16-00607-f003:**
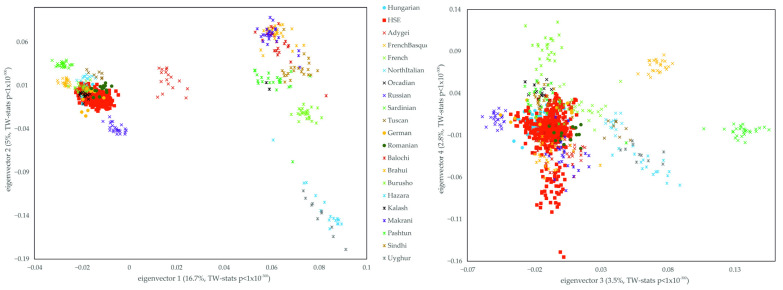
PCA analysis of the HSE group and Hungarian, HGDP Eurasian and Estonian data-based reference populations. Eigenvalues of eigenvectors 1, 2, 3 and 4 were 11.053, 3.299, 2.320 and 1.844, respectively. The statistical significance and the explained variance of each principal components are provided in Appendix A. Each symbol represents one individual.

**Figure 4 genes-16-00607-f004:**
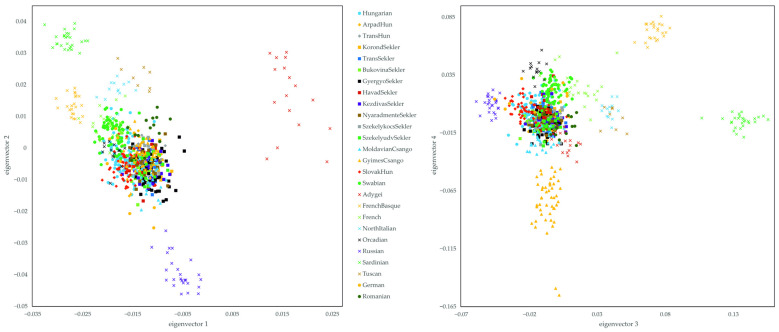
PCA analysis of the separated HSE populations and Hungarian, HGDP European and Estonian data-based reference populations. Eigenvalues of the eigenvectors are the same as those described in Figure 2. The statistical significance and the explained variance of each principal components are provided in Appendix A. Each symbol represents one individual.

**Figure 5 genes-16-00607-f005:**
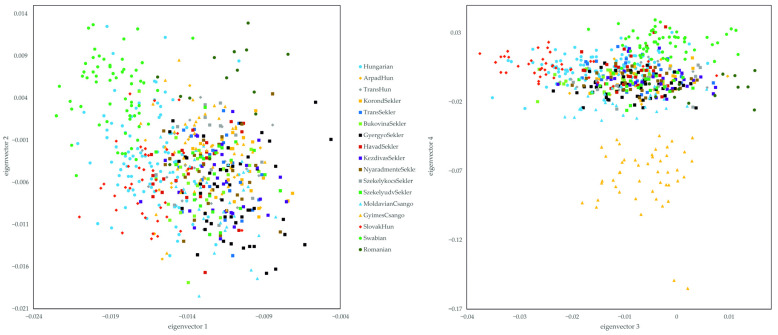
PCA analysis of the separated HSE populations and Hungarian and Estonian data-based Romanian reference populations. Eigenvalues of the eigenvectors are the same as those described in Figure 2. Note that this PCA is identical to the previous analysis, except that this is an even more restrictive representation of that. The statistical significance and the explained variance of each principal components are provided in Appendix A. Each symbol represents an individual.

**Figure 6 genes-16-00607-f006:**
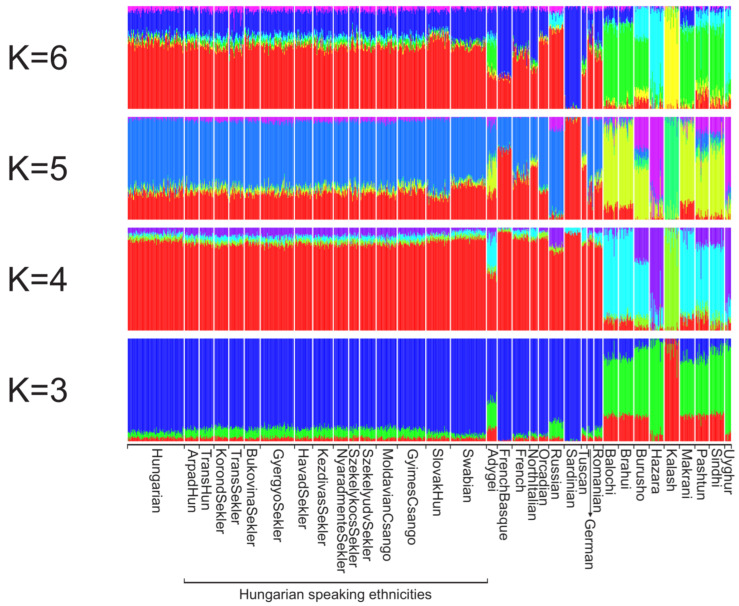
ADMIXTURE analysis results of the separated HSE populations and Hungarian, HGDP Eurasian and Estonian-based reference populations with K = 3–6. Cross-validation error was the lowest applying four clusters. Cross-validation error values at K = 3, K = 4, K = 5 and K = 6 was 0.57774, 0.57735, 0.57745 and 0.57795, respectively. Each column represents one individual and each column group represents a population. Appendix A indicates all ADMIXTURE analysis results from K = 2 to K = 10.

**Figure 7 genes-16-00607-f007:**
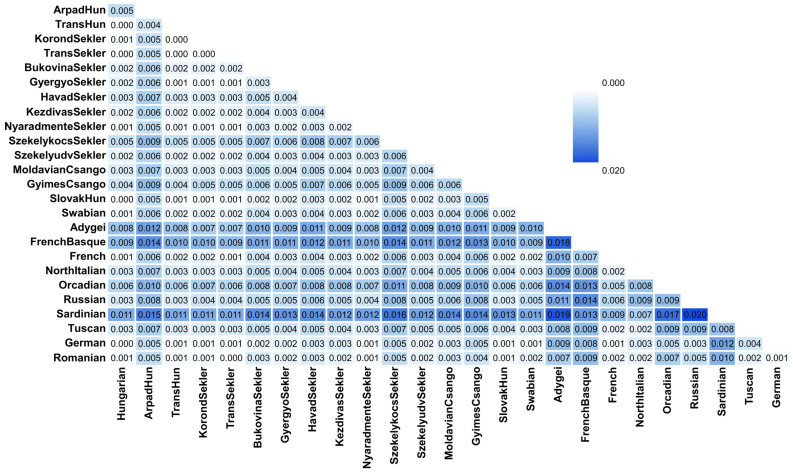
F_ST_ matrix calculated by the SMARTPCA software showing the average pairwise allele frequency differentiations between the investigated HSE populations and Hungarian, HGDP European and Estonian data-based reference populations. Appendix A contains the F_ST_ results of all investigated populations, while Appendix A indicates the standard errors of the F_ST_ calculations.

**Figure 8 genes-16-00607-f008:**
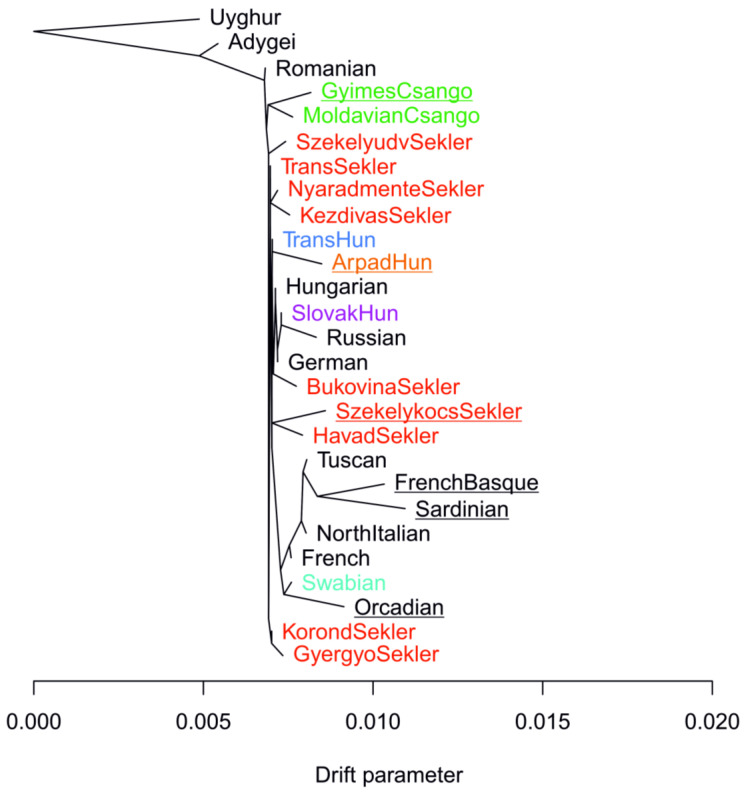
TreeMix analysis results. The calculated maximum likelihood tree. Target populations are highlighted by color, and populations deviating from the Central European drift pattern are underlined. Appendix A shows the residual fit of the TreeMix run.

**Figure 9 genes-16-00607-f009:**
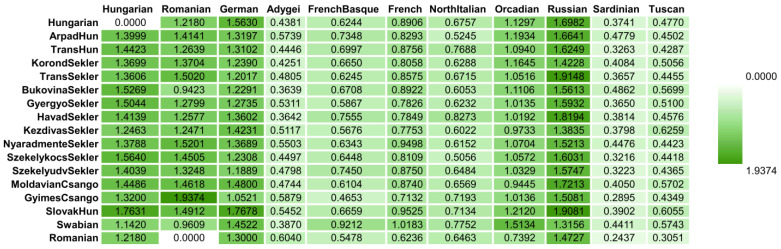
Identity-by-descent (IBD) segment analysis of the investigated HSE populations and Hungarian, HGDP European and Estonian data-based reference populations.

**Figure 10 genes-16-00607-f010:**
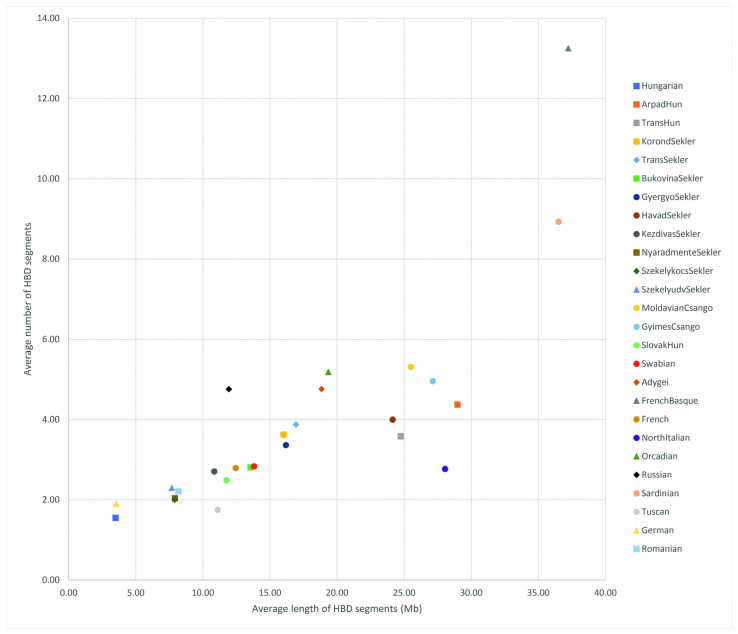
Genome-wide average autozygosity of the investigated HSE populations and Hungarian, HGDP European and Estonian data-based reference populations.

**Table 1 genes-16-00607-t001:** Notable D-statistics reflecting genetic affinity patterns among tested populations.

W Population 1	X Population 2	Y Population 3	Z Population 4	D-Statistics	Z-Score
Yoruba	Hungarian	Romanian	ArpadHun	0.000303	2.316
Yoruba	Hungarian	Romanian	BukovinaSekler	0.000262	2.122
Yoruba	Hungarian	Romanian	HavadSekler	0.000266	2.165
Yoruba	MoldavianCsango	Romanian	ArpadHun	0.000323	2.445
Yoruba	MoldavianCsango	Romanian	GyimesCsango	0.000278	2.405
Yoruba	Hungarian	German	Swabian	0.000073	0.609

## Data Availability

All data generated or analyzed during this study are included in this published article and its Appendix A. Some of the datasets are available in public online repositories. The HGDP data are available directly from the homepage of Rosenberg lab at the Stanford University (https://rosenberglab.stanford.edu/hgdpsnpDownload.html; accessed on 28 October 2024), while the 1KGP (https://www.internationalgenome.org/category/vcf/; accessed on 28 October 2024) data are available directly from their ftp server (https://ftp.1000genomes.ebi.ac.uk/vol1/ftp/release/20130502/; accessed on 28 October 2024). Populations of the Estonian Biocentre can be downloaded from their repository (https://evolbio.ut.ee/; accessed on 28 October 2024). The Csango, Sekler, Hungarian, Swabian and SlovakHun datasets, according to the Hungarian Human Genetics Act 2008/XXI, cannot be uploaded to a public online database, but can be obtained upon reasonable request via e-mail from the corresponding authors.

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
