# Peer review of "Comparative Analysis of the Genetic Composition of Minorities in the Carpathian Basin Through Genome-Wide Autosomal Data"

_genes, 2025, doi:10.3390/genes16050607_

Round 1

Reviewer 1 Report

Comments and Suggestions for Authors

Summary

This manuscript, Comparative analysis of the genetic composition of minorities in the Carpathian Basin through genome-wide autosomal data, presents genome-wide autosomal SNP data from several previously understudied Hungarian-speaking ethnic minorities. Using principal component analysis, ADMIXTURE, TreeMix, and IBD/HBD segment analyses, the authors investigate fine-scale genetic structure, admixture patterns, and historical isolation among groups in the Carpathian/Pannonian Basin. The study leverages a well-curated and novel sample set along with comparative datasets from publicly available cohorts. The major findings highlight overall genetic homogeneity among Hungarian-speaking populations, with distinct genetic signatures preserved in isolated groups such as the Csangos and Danube Swabians. The topic is timely and valuable, contributing to both historical and medical genetic understanding of regional variation. However, there are important issues regarding analytical depth and interpretation of results that should be addressed.

Major Comments

Lack of novel methodological approaches or modeling depth
While the study presents a substantial and valuable genome-wide dataset from underrepresented populations, the analytical framework remains relatively basic by contemporary standards. The investigation relies primarily on PCA, ADMIXTURE, TreeMix, and IBD/HBD sharing, all of which are established tools but have known limitations when used alone to infer fine-scale structure or historical admixture events.

Given the subtle genetic differences expected among neighboring Central and Eastern European populations, especially those with shared language and culture, the use of more advanced statistical methods would have been appropriate and is increasingly expected in the field. For example:

  • f3-statistics to formally test for admixture events
  • f4-statistics or D-statistics to more robustly test hypotheses about specific population relationships 
  • qpAdm modeling to estimate admixture proportions between proposed source populations
  • qpGraph to fit explicit models of population relationships, evaluating drift, admixture, and divergence events 

Without these or similar formal tests, the study relies heavily on descriptive, visual patterns (e.g., slight PCA separations) to infer historical events, which is prone to subjective interpretation. The authors should incorporate at least some additional formal analyses to strengthen their conclusions. If they choose not to add these methods, the discussion should be more cautious, stating that their findings are primarily exploratory.

Minor Comments

  • For ethical reasons, I recommend using terms like consanguinity and endogamy in place of 'inbreeding'
  • "Hungarian-derived ethnicities" and "Hungarian-speaking ethnicities" are used interchangeably but may imply different meanings
  • The PCA plots are difficult to interpret due to overlapping points and color choices- I would especially appreciate a consistent shape or other distinguishing feature to make it easier to identify the points that represent novel sequencing data derived in this study in Figures 2, 3, and 4
  • Similarly, the Treemix figure would benefit from some kind of color coding that clearly indicates the populations that are novel in this study
  • Several minor grammatical issues (e.g., "regarding to modern state boundaries," "orientiation" instead of "orientation")  
  • The authors cite their prior work on Csango Turkic ancestry in the Discussion but could better integrate these (interesting) findings into the main text 

Reviewer 2 Report

Comments and Suggestions for Authors

The manuscript presents an extensive genome-wide autosomal SNP analysis of Hungarian-speaking ethnic groups in the Carpathian Basin, utilizing modern population genetics tools. It is a well-designed and potentially impactful contribution to European population genetics, offering new insights into regional structure and genetic isolation among minorities that are underrepresented in genomic research.

This study represents a novel and significant contribution to the field of population genetics, particularly focusing on underrepresented Hungarian-speaking ethnic minorities using genome-wide autosomal SNP data. The exploration of genetic isolation and ancestry patterns in minorities is unique and relevant. The manuscript is well-structured and logically organized. Each section transitions smoothly from historical context to study rationale. However, while the overall scientific quality is commendable, there are some points that require refinement.

Here are my detailed comments and suggestions:

  1. Although standard tools are used, the manuscript lacks some technical parameter settings, data pruning thresholds, and rationale for analytical decisions.
  2. No information on Hardy-Weinberg equilibrium filtering, minor allele frequency thresholds, or per-individual call rates
  3. The reliance on self-declared ethnicity and three-generation pedigree reports needs validation.
  4. No mention is made of independent checks (e.g., mitochondrial/Y-haplogroup confirmation) to support homogeneity.
  5. The exclusion of key populations such as Hungarian minorities in Serbia is weakly justified. This exclusion limits the regional scope and reduces the comprehensiveness of the analysis.
  6. It is suggested to include more discussion of ancestral component assignment, particularly when interpreting slight differences as evidence of unique ancestry.
  7. Adding details on statistical analysis like confidence intervals, permutation tests, cross-validation rationale is suggested.
  8. The authors interpret the study findings to personalized medicine and pharmacogenetics, but these claims are largely speculative.
  9. No direct association between identified genomic traits and medical or pharmacogenetic markers is presented.
  10. Some PCA plots are cluttered and difficult to interpret; separate panels for key findings could improve clarity.

Overall, this manuscript has strong potential and contains substantial novel data, but the authors must address the outlined key areas needing improvement.

Round 2

Reviewer 1 Report

Comments and Suggestions for Authors

I appreciate the authors’ thoughtful revisions. They have adequately addressed all of my previous concerns, and I am now satisfied with the manuscript in its current form. I have no further suggestions and support its publication.